# Does Mobile Internet Use Affect the Depression of Young Chinese Adults? An Instrumental Variable Quantile Analysis

**DOI:** 10.3390/ijerph19084473

**Published:** 2022-04-08

**Authors:** Yuyan Chen, Lin Wu, Zenghua Guo

**Affiliations:** 1School of Sociology, Wuhan University, Wuhan 430072, China; chenyuyan@whu.edu.cn (Y.C.); wlin@whu.edu.cn (L.W.); 2School of Marxism, Hubei University of Economics, Wuhan 430205, China

**Keywords:** depression, mobile internet use, quantile regression

## Abstract

Background: With the advancement of the digital age, the links between mobile Internet use (MIU) and mental health have attracted the attention of scholars. This paper focuses on the relationship between MIU and depression across the entire distribution of young adults’ depression. Methods: Based on nationally representative data from the 2018 China Family Panel Studies (CFPS), we explore whether and to what extent MIU affects depression in young adults in China through instrumental variables. In addition, we employ a quantile regression approach to explore the depression–MIU gradients and examine potential mediation mechanisms by exploiting variation in several potential intermediates available. Results: 2SLS estimate suggests that MIU is associated with a decrease in young adults’ depression by 1.526 points. Besides, the effect of MIU was only significantly negative at the 0.8 to 0.96 quantiles. Discussions: MIU reduces the level of depression in people with higher levels of depression, older age, and who use the Internet for communicative purposes. However, there is no significant gender difference in MIU. In addition, young people will improve their feeling of social comparison when using mobile Internet, thus making them less depressed. Conclusions: MIU has a significant positive impact on depression among young Chinese adults.

## 1. Introduction

According to data from the China Health Commission, up to 54 million people in China suffer from depression, accounting for 4.2% of the total population. The earliest onset of all types of depression is around the age of 14, and the prevalence is three times higher in individuals aged 18–29 than those aged 60 and older [1]. According to the “Statistical Report on Internet Development in China [2]” released by the China Internet Network Information Center (CNNIC), the number of Internet users reached 1.011 billion, and the Internet penetration rate reached 71.6%. Among them, the proportion of mobile Internet access reached 99.6%. Compared with the data of previous years [3], all indicators have maintained steady growth, indicating that the mobile Internet has become an indispensable and important tool in modern life. People use the Internet to establish new social connections, search for information, and conduct business activities. At the same time, the risk of the Internet eroding offline life is also increasing. A series of problems, such as interpersonal isolation caused by addiction to the Internet and loss of self-esteem caused by comparison, threaten the mental health of Internet users. The academic community has launched a discussion on the issue of the impact of the Internet on mental health.

For a long time, people have had both positive and negative views on Internet use. Opponents believe that an increasing proportion of young adults with psychotic disorders is related to inappropriate use of the Internet. Improper Internet use will lead to a variety of pathological characteristics such as increased dependence, social withdrawal, guilt and difficulty in inhibiting Internet desire, and further lead to depressive disorder, anxiety disorder as well as depression and anxiety disorder (comorbid disorder). Luo et al. [4] tested 75 adolescents who met the DSM-IV standards. They found that most Internet-addicted adolescents have comorbidities of depression and anxiety, requiring both antidepressants and antianxiety drugs for psychotherapy. Internet is free from the constraints of time and space and has the advantages of maintaining a low cost of cross-border communication. Therefore, netizens, who conduct online activities through computers and the Internet, shift from intuitive, personality-dependent face-to-face communication to fragmented, text-based visual platform communication. The weak relationship with strangers has impacted their strong relationship with close people, often resulting in a withered communication situation, misunderstandings, negative emotions, and other adverse consequences [5,6]. Proponents suggest that human–computer interaction only requires low-cost hardware equipment and convenient software facilities. Internet connection can create more “random encounters” than offline communication, initiating a weak social relationship, so that individuals without special preparations have more opportunities to connect with other people. It is more convenient to create networked multi-point individual linkages or networks cluster behavior, improving subjective well-being, sense of gain and suppressing depression [7,8].

Research on depression issues has potential societal benefits. Data show that the prevalence of depression in Chinese adults is relatively high [9], the proportion of depression is increasing year by year, and the symptoms are getting worse. The novel coronavirus epidemic has exacerbated this trend [10]. In developing countries, most people lack understanding of depression, cannot effectively identify symptoms, and patients cannot access psychiatric treatment in a timely manner [11]. In addition, the research on youth mental health in China started relatively late, and there is still a lack of systematic evaluation systems, effective intervention measures, and large-scale survey results [12]. The existing academic achievements on the impact of MIU on depression have achieved complex results, but most scholars focus on the elderly and adolescents, rather than the youth. This article attempts to show the mental health status of young people through statistical data, exploring how the Internet and its different use purposes have emotional effects. We attempt to use an instrumental variable approach to explore the causal association and the underlying mechanisms between mobile Internet use (MIU) and depression in young adults (aged 18–40 years). On the one hand, this paper focuses on the mental health problems of young people, especially the specific topic of depression, and contributes empirical experience to the current field. On the other hand, in view of the increasingly serious situation of depression in China, we attempt to provide possible policy recommendations in the field of mental health, to provide a reference for actually solving young adults’ depression and reducing the prevalence of depression.

## 2. Prior Literature

Depression is referred to as a perceived pessimistic mood [13] or academically defined as feelings of helplessness and hopelessness [14], the level of feelings of guilt and worthlessness [15], or psychomotor retardation [16]. The ICD-11 and DSM-5 have basically the same diagnostic description of depressive episodes, including depressive mood and loss of interest, accompanied by other cognitive, behavioral, or autonomic symptoms, which have a significant impact on the level of individual social function [17,18]. Young people’s emotional management ability is relatively immature, but they are at the center of the vortex of fierce social competition. They are most likely to be impacted by the loss of competition and face the challenge of re-adaptation. Being accepted and new social engagement are goals young people strive for. In the context of a competitive society, understanding and studying depression among young adults is a key priority, mainly because depression and health are closely linked. Depressive symptoms can negatively affect a person’s mood, thinking, self-feeling, interpersonal communication, and physical function [19], leading to social isolation and even an increase in the level of suicidal ideation [20]. At present, the number of common mental disorders and psychological problems in China is increasing year by year, and malignant cases caused by personal extreme emotions occur from time to time. It is harmful to social stability, interpersonal harmony, and public happiness. Given the importance of depression among young adults, its determinants have become the focus of a large body of studies [21,22].

A growing number of studies closely related to our work explore the role of Internet use in influencing the depression of young people [23,24]. Extant studies on the relationship between Internet use and depression in adolescents have yielded mixed results. For example, employing a nationally representative sample of 1787 young adults ages 19–32 years in the U.S., Lin et al. [25] demonstrate that there is a significant association between Internet use and depression. Maladaptive patterns of Internet use may result in mood dysregulation. Using data from a longitudinal study of young adults by Growth from Knowledge (GfK) (GfK KnowledgePanel^®^, 2013), Shensa et al. [26] also suggest that maladaptive use patterns may explain the association between Internet use and depression. The work by Jelenchick et al. [27], however, suggests that Internet use has no significant impact on depression in older adolescents. Interestingly, a paper by Selfhout et al. [28], documents that Internet use may not increase depression when they are addicted to the Internet for communication purposes.

Despite the fact that the incidence of young adults’ depression is increasing in China, there is little work that empirically analyzes the association between Internet use and depression in the context of China. Specifically, most of the relevant research has focused on the impact of social networks (e.g., self-disclosure, personal characteristics, substance use, life satisfaction) [29,30,31,32]. For instance, using evidence from the online big data platform, Yu et al. [11] document that the Internet is a proper source of help for people with depression. One exception is the recent work by Huang [33], employing meta-analysis and using a sample of 19,652 participants, who find that the correlation between Internet use and depression is not significant.

In general, most existing studies investigate the association between Internet use and depression among young adults. However, these studies have three limitations. First, while growing research has focused on the relationship between Internet use and depression among young adults, there is no empirical work analyzing the effect of Internet use across the distribution of young adults’ depression. Second, possibly due to limitations of data analysis, most previous work does not provide a good solution to address endogeneity problems, implying that their conclusions cannot be interpreted as causal effects. Third, although existing research has suggested Internet use plays a role in affecting young adults’ depression, few studies empirically explore the potential mechanisms behind these impacts.

## 3. Data and Methods

### 3.1. Data and Variable Selection

This paper uses data from the Chinese Family Panel Studies (CFPS), a nationally representative longitudinal survey funded by Peking University and the National Natural Science Foundation of China and initiated by the Institute of Social Sciences of Peking University in 2010. In 2008 and 2009, the CFPS conducted initial and follow-up surveys in Beijing, Shanghai, and Guangdong, then officially conducted investigations in 2010. All baseline family members defined by the 2010 baseline survey and their future consanguineous/adopted children will be regarded as genetic members of CFPS and become permanent tracking objects. The fifth round of the national survey (CFPS 2018) began on 5 June 2018 and lasted until May 2019. The sample covers 25 provinces/cities/autonomous regions in China, including about 95% of the population. It adopts implicit stratification, multi-stage, multi-level probability proportional to size (PPS). Therefore, the sample of CFPS can be regarded as a national representative sample.

In order to investigate the factor of MIU, we selected 2018 data for statistical analysis. Specifically, a total of 33,326 people were interviewed in the 2018 CFPS survey, and the samples mainly used in this paper came from the household economic database and personal database. According to research needs, we focused on the youth group between 18 and 40 years old and excluded those samples lacking information on core variables. The research used 10,499 observations after data cleaning to carry out the analysis.

The dependent variable of the study was the individual’s level of depression, measured by using a 12-item version of the Center for Epidemiologic Studies of Depression Scale (CES-D) questionnaire. It is measured on a 51-point scale ranging from 22 to 72, with higher scores indicating higher levels of depression. Figure 1 below shows the distribution of observed depression levels. The distribution of depression levels in young adults is skewed to the left, with mild depression accounting for the largest proportion, and severe depression being relatively rare.

The independent variable is MIU. MIU is assessed by the question “If the respondent uses mobile Internet through any mobile device (such as smart phone and iPad) in daily life” (1 = yes, 0 = no). Based on the available literature, we controlled for a range of demographic variables that might be associated with MIU and young adults’ depression. Specifically, they include the squared term of age, gender (1 = male, 0 = female), current marital status (1 = married, 0 = other), place of residence (1 = rural area, 0 = urban area), employment status (1 = Working or in school, 0 = neither), and perceived wealth gap (1 = there is a serious wealth gap in China, 0 = there is no serious wealth gap in China). In addition, we included the provincial dummy variable in the control.

To control for potential endogeneity, this study used two instrumental variables, attitudes towards the Internet and household telecommunication expenses. The first instrumental variable “attitude to the Internet” corresponds to the “importance of the Internet for you to obtain information” in the questionnaire, and has been re-coded to form a corresponding binary variable (1 = important, 0 = other).

Young people who have a positive attitude towards the Internet will correspondingly increase the frequency and duration of their use of mobile Internet. However, it did not directly affect depression in young people. The second instrumental variable is household telecommunication expenses, which measures the monthly mailing and communication expenses of the households where the sample is located, including telephone, mobile phone, Internet access, mailing and other expenses. We believe that the household telecommunication expenses are not directly related to the dependent variable. We obtain data from the family database, and then correspond them to individuals through family number matching. There is a positive correlation between household telecommunication expenses and the use of mobile Internet. Table 1 presents the definitions and summary statistics of the dependent and independent variables.

### 3.2. Empirical Strategy

To investigate the effect of MIU on the depression among young adults in China, we first use ordinary least squares (OLS) and two-stage least squares (2SLS) regressions to estimate the following model:Dep_i_ = α_0_ + α_1_MIU_i_ + φX_i_ + ε_i_(1)
where the dependent variable is depression of individual i, the independent variable is MIU status, X_i_ is a vector of control variables, including the squared term of age, gender, current marital status, place of residence, employment, perceived wealth gap, provincial dummy variable and ε_i_ is the error term.

Since the effect of MIU may differ across the depression distribution, and MIU may be endogenous, we utilize the quantile regression approach proposed by Koenker and Bassett [34] to explore the effect of MIU on the distribution of depression among young adults:

First-stage Estimate:MIU = β_0_ + β_1_Z_i_ + β_2_X_i_ + η_i_(2)

Second-stage Estimate:Dep_i_ = α_0_ + α_1_(Estimate MIU_i_) + α_2_X_i_ + ζ_i_(3)
where Z_i_ is a vector of instruments that are correlated with MIU_i_. We use two IVs—attitude towards the Internet and household telecommunication expenses. We adopt two ways to test that the two instruments use as valid instruments: First, we treat the two instruments as control variables to analyze whether there is a direct effect of household telecommunication expenses and attitude towards the Internet on depression. Second, we use overidentification tests to examine whether the two instruments meet the need for instrument exogeneity.

In general, the article mainly adopts the following five strategies. First, using instrumental variable regression solves the endogeneity problem of the independent variable, and compares with the results of ordinary least squares regression; second, using heterogeneity analysis investigates the difference in the influence of the mobile Internet on depression levels; third, through quantile regression, we describe and analyze how mobile Internet use affects depression at each quantile; fourth, we conduct robustness tests by replacing data samples and refining independent variable categories; fifth, we explore the potential mechanisms of MIU affecting depression.

## 4. Results

### 4.1. Correlates of Depression among Young Adults in China

Table 2 presents the correlation matrix of the variables used in this paper. The data show that there is a significant negative correlation between MIU, gender, employment status and depression, while age and place of residence show a significant positive correlation with depression. In other words, when MIU increased, depression decreased; when age increased, depression increased. Compared with female and non-school or non-working individuals, male and school or working populations have lower depression, while individuals in rural areas have higher levels of depression than individuals in urban areas.

We start by exploring the correlates of young adults’ depression based on the OLS regressions (see Table 3). We first regress depression on MIU without including other covariates (column 1). Next, we report the estimated MIU effect on young adults’ depression, conditional on other covariates and provincial dummies (column 2 of Table 3). The results suggest that even after controlling for covariates, the positive effect of MIU on depression in young adults remained significant. Specifically, the depression of young people who use mobile internet is lower than those of young people who do not use mobile internet.

With respect to other covariates, our estimates indicate that men have lower depression levels than women; married people have lower levels of depression than other groups, and young adults who are employed or in school have lower levels of depression than unemployed and school leavers. On the other hand, older youth have higher levels of depression; rural youth have higher levels of depression than urban youth, and youth who perceive a severe wealth gap have higher levels of depression.

### 4.2. Impacts of MIU on Depression and Its Distribution

To solve possible endogeneity issues, we further use household telecommunication charges and attitudes towards the Internet as IVs for MIU. Before using the instrument variable (IV) approach, we regress depression on the two instruments to explore whether household telecommunication charges and attitude towards the Internet exert a direct effect on young adults’ depression. The variance inflation factors (VIF) are calculated to premeditate the potential multicollinearity between the main metrics.

As shown in the data in columns 1 and 2 of Table 4, instrumental variables did not directly affect the level of depression in young adults. Columns 3 and 4 show the regression results of the first stage of 2SLS (two-stage least squares), and both instrumental variables have significant effects on the core independent variable. At the same time, the value of the F statistic is 351.36 (*p* < 0.01), rejecting the null hypothesis of the weak instrument, and the instrumental variable is valid. The value of the variance inflation factor (VID = F) of key variables is not greater than 2, which indicates that there is no multicollinearity problem between other variables except the provincial dummies. Finally, through the endogenous test, we can consider that both instrumental variables are exogenous, and the results of the instrumental variable regression are consistent with the results of the OLS regression model, indicating the rationality of the instrumental variable method.

After taking the endogeneity of MIU into account, the effect of MIU on young adults’ depression remains significantly negative (see Table 4), indicating MIU significantly reduces depression in young adults. Our 2SLS estimate suggests that MIU is associated with a decrease in young adults’ depression by 1.526 points. This represents a decrease of 4.78 percent. The instrumental variable regression showed the average effect of MIU on the depression of young people (see Figure 2), and the quantile regression method was introduced in this study to explore the specific distribution of this effect. The results showed that the mean effects derived from 2SLS (two-stage least squares) regression masked a considerable degree of heterogeneity (see Table 5). Specifically, the effect of MIU was only significantly negative at the 0.8 to 0.96 quantiles (see Table 6). This suggests that young adults with higher depression benefited more from MIU after controlling for covariates.

To better understand the impact of MIU on young adults’ depression, we next examine whether the impact of MIU differs by age, gender, or purpose of use. This paper divides the existing sample into two groups according to age; one group is 18 to 28 years old, and the other group is 29 to 40 years old. MIU has a significant effect on reducing depression in older people (29–40 years old), but not in younger age groups (18–28 years old) (see Table 7).

In China, the effect of MIU on depression in males and females was not statistically significant, meaning that there was no gender difference (see Table 8). Through the two-sample *t*-test, we found that there are differences in the preference of the two genders for using the Internet. Females are more likely to engage in online work and online business activities, and males are more inclined to use the Internet for entertainment. There is no significant difference in online social behavior between males and females.

The effect of mobile Internet access on depressive mood varies according to the purpose of use. The four items of the data correspond to “How often do you use the Internet (including the intranet of your company) for work?”, “Do you use the Internet for social interaction (such as chatting, posting Weibo, etc.)”, “How often do you use the Internet for entertainment (such as watching videos, downloading songs, etc.)?” and “How often do you use the Internet for business activities (such as using online banking, online shopping)?”. Based on the existing literature, we classify social interaction as “communicative use” for the purpose of interpersonal communication, and work, entertainment, and business activities as “non-communicative use”. Data analysis shows mixed results (see Table 9). Higher frequency of using the Internet for social or recreational activities can significantly reduce the depression level of youth, and the influence factor of social interaction is greater than that of recreational activities. Higher use of the Internet for work or business activities did not affect depression in youth.

### 4.3. Sensitivity Analysis

In order to examine the robustness of the results, this paper will conduct a series of robustness tests from two perspectives. First, we extend the range of age from 16 to 40 years old. The benchmark regression results are shown in the table below, the use of mobile Internet still has a significant negative impact on the depression level of young people (see Table 10). Similar results were obtained in the instrumental variable analysis (see Table 11).

Second, we replaced “use of mobile Internet” with “use of stationary computer” as an independent variable in the regression. The results of benchmark regression and instrumental variable regression showed that the use of a computer to surf the Internet had a significant negative effect on the depression level of young people (see Table 12).

### 4.4. Mechanisms

To offer more insight into the nexus between MIU and depression among young Chinese adults, we further explore the mechanisms through which MIU may lead to a lower level of depression by exploiting variation in several potential intermediates available in our dataset. Specifically, we have data on subjective well-being, interpersonal self-assessment, subjective social status, and subjective economic status. Subjective well-being is measured on a 10-point scale based on the response to the question “How happy do you feel?”, with 1 indicating “very unhappy” and 10 indicating “very happy”. The measure of interpersonal self-assessment stems from a question that asks how the respondents evaluate their personal relationships, indicating with 1 “very bad” and 10 indicating “very well”. Subjective social status and economic status are measured on a 5-point scale based on the question “How do you rate your local social status” and “How do you rate your income in the local area”, respectively.

Summary statistics on the intermediates are presented in Table 13. The two-sample *t*-test results suggest young people who use mobile Internet have lower perceptions of their subjective social status and income status than those who do not use mobile Internet. However, the former has higher happiness and interpersonal self-evaluation than the latter.

Through benchmark regression, while controlling for covariates it can be found that MIU has a significant positive impact on subjective well-being and self-evaluation of interpersonal relationships, however, MIU has a significant negative impact on both subjective social status and subjective economic status. These four paths may all be potential mechanisms by which the independent variable affects the dependent variable (see Table 14). Next, we calculated the effect of the four mediator variables and the values of each path and performed the bootstrap test. Among them, the two variables of subjective well-being and self-evaluation of interpersonal relationships cannot pass the bootstrap test, so we can think that Internet use does not reduce the level of depression by improving the subjectively perceived well-being or interpersonal relationships of young people. Therefore, in the final discussion, only the remaining two mediator variables are kept.

The mediating effect of subjective social status and subjective economic status is shown in the figure below. Column 1 of Table 15 shows the effect of mobile Internet use on depression levels after controlling for covariates, and the values are directly derived from the previous instrumental variable regression results. Columns 2 and 3 of Table 15 are the estimated mediation effects calculated by taking the two variables of subjective social status and subjective economic status into control, respectively. When subjective social status is used as the mediating variable, the result of direct effect is −0.487, which is significant at 0.05 level; the result of indirect effect is −0.064, which is significant at 0.01 level. When subjective economic status is used as the mediating variable, the result of direct effect is −0.466, which is significant at 0.05 level; the result of indirect effect is 0.078, which is significant at 0.01 level. The data show that both mediator variables had significant effects on depression levels, and after the mediator variables were included in the control, the effect of mobile internet use remained significant, but the estimated coefficients contracted from −1.41 to −0.49 and −0.47. This shows that the influence of mobile Internet use on the depression level of young people is partially mediated by their subjective social status and subjective economic status.

The data show that estimates of direct effects are positive and indirect effects are negative, with opposite signs (see Figure 3). According to the research of Wen et al. [35], if the signs of ab and c′ are opposite, it needs to be explained according to the masking effect, and finally report the value of |ab/c′|. When subjective social status was used as the mediating variable, the indirect effect accounted for 13.05% of the direct effect; when the subjective economic status was used as the mediating variable, the indirect effect accounted for 16.70% of the direct effect. The masking effect is similar to the mediating mechanism, but the effect is the opposite. The appearance of the masking effect means that the mediator variable masks the influence of the independent variable (whether to surf the Internet) on the dependent variable (depression) to a certain extent. In this study, although the overall effect of MIU reduces depression in young adults, this particular mediating pathway produced the opposite effect. Specifically, MIU reduced the self-assessment of social status and economic status of young people, which had aggravated the degree of depression. However, fortunately, there are other more significant influencing factors related to the total effect, so that MIU decreases the depression of young people finally.

## 5. Discussion

Previous studies on the impact of MIU on depression mostly concentrate on the elderly and adolescents, and the attention on youth is relatively lacking. At the same time, the existing academic achievements have not focused on the distribution of the impact of MIU on young people, and have not been able to solve the endogenous problem. Using data from the highly representative and reliable China Family Tracker Survey (CFPS), this study explores how MIU affects depression among young Chinese and examines the effect at different quantiles of depression symptomatology. At the same time, the article examines the age differences, gender differences and differences in the purpose of MIU to reduce depression, and explores the potential mediating mechanism.

Through OLS regression, instrumental variable regression and their tests, we believe that young people who engage in MIU show lower levels of depression than those who do not. Quantile regression showed the difference in the impact of MIU on groups with different levels of depression. The results showed that MIU only had a significant positive effect on a subset of young people with the highest levels of depression. We speculate that people with lower levels of depression have achieved sufficient happiness through daily interpersonal communication and entertainment projects. On the contrary, relatively more depressed groups tend to self-isolate in real life and stay in a lonely environment for a long time, so they are more dependent on the Internet and pay more attention to the functionality of the Internet [36,37,38]. Both the use of computers and mobile devices to surf the Internet can significantly make young people less depressed and feel relief. The utility level of mobile surfing (−1.526) is slightly higher than that of using computers to surf the Internet (−1.448). It is higher than the ratio of computer Internet access (the maximum is the use of desktop computers to access the Internet, accounting for 34.6%).

We find that while MIU is associated with decreased depressive feelings in young adults, the beneficial impacts of MIU on depression are greater for those aged 29–40 years when they use the Internet for communication purposes. This could be due to the fact that as they grow older, these adults will be busy with work and bear more family responsibilities, free leisure time will be reduced and social communication will be reduced accordingly. Internet use makes up for their social support, thereby improving their mood. The gender difference in the influence of MIU is not significant, which could be due to the fact that the parenting style of many one-child families in China has changed the traditional gender role expectations. Many females are not only good at interpersonal communication, but also keen on Internet surfing and online games, showing masculinity. Furthermore, some males also show more femininity.

Using data from the nationally representative surveys of U.S. adolescents in grades 8 through 12 (N = 506,820), Twenge et al. [23] document that the use of the Internet is more likely to increase depressive symptoms only among girls, because the Internet may have larger effects on adolescent girls’ mood than on boys’. Mars et al. [22] also find that high levels of internet use are more likely to increase the prevalence of depression among females. There is no gender difference in the effect of Internet use on the depression level of young people, but the purpose of using the Internet is different for both sexes. Women are more engaged in online work and online business activities, and men are more likely to use the Internet for entertainment; there is no significant difference in online social behavior between the sexes. These results are consistent with studies by scholars such as [20,39], but contradict many claims that women are more likely to become addicted, or have lower self-efficacy, and perceive greater anxiety and depressive mood [40,41].

The notion that women report lower levels of self-esteem, self-efficacy, etc. through Internet use generally comes from earlier research. In the early days of Internet popularization, men were more likely than women to have access to Internet devices such as computers, and the level of application operation was generally higher than that of women. However, in the context of the popularization of the Internet, the digital literacy of netizens has been improved in an all-round way, and the gender difference in device application ability has narrowed to an unrecognizable level. From this, we speculate that many one-child families in China have changed the way their offspring are reared, surpassing traditional gender role expectations, and thus the gender distinction of offspring has been weakened. Both men and women have developed sufficient communication skills and common interests. Many women are not only good at dealing with online work, but also keen on surfing the Internet and playing online games, showing a tendency consistent with early male Internet users. Furthermore, some men also showed more female preference.

In addition, some scholars believe that the root of Internet addiction lies in the purpose of using the Internet rather than gender itself [20]. Research data also show that different genders have different Internet usage preferences. A possible explanation path is that in the digital age, society’s dependence on the Internet has gradually deepened. The Internet has increasingly become the first choice for people to retrieve information, communicate, and shop. Therefore, people use some of the basic functions of the Internet not out of preference, but to make choices that conform to the development of the times.

Overall, the use of mobile internet for social purposes was more likely to reduce depression levels in young adults. However, interestingly, the data show that online entertainment, a non-communicative activity, can also alleviate depression, although its impact factor is slightly smaller than that of online social interaction. The likely reason is that there is a difference between legitimate online entertainment and addictive online entertainment. Scholars’ previous research conclusions are mostly about how online behaviors that have evolved into pathological use aggravate depressive symptoms [42,43]. We believe that moderate use of the Internet for entertainment provides youth with social-emotional support and therefore reduces users’ depression levels. This is also the difference between online entertainment and online work and online business activities.

Young people reported lower subjective economic status and subjective social status by using the mobile internet, but this result also significantly reduced depression levels. We believe that the decline of subjective social status and subjective economic status is related to social comparison. People could obtain social networking profiles presented by others on Internet platforms. Compared with their own situation, the inevitable comparisons erode their self-esteem [7]. The existing literature has proved that using the Internet will lead to more upward social comparisons, thereby reducing self-evaluation [5,44,45], This is consistent with some of the findings of this study. One conjecture is that young people have a strong incentive to succeed. Even if their self-perception is temporarily lowered due to social comparison, they may see the comparison object as a goal, or even an expected future achievement, thereby boosting self-confidence and self-motivation and offsetting negative emotions. Another explanation is that the rich network expressions show the multi-faceted lives of different strata groups, making social comparisons in various directions possible. After experiencing upward comparisons, young people actively adopt avoidance strategies and turn to more downward social comparisons, thereby restoring good self-perception, relieving anxiety, and reducing depression.

In addition, we believe that with the popularity of the Internet, more people are generally more inclined to appear positive on the Internet. This shift can help netizens in turn understand other people’s displays and identify confounding factors, thereby avoiding comparisons when they perceive upward social comparisons. Today, on the Internet platform, there is a commercial behavior of showing off wealth in exchange for traffic value, which undoubtedly provides a way for audiences to avoid upward social comparisons. When the audience believes that the performance component is greater than the real component, it will deconstruct this illusory wealth and eliminate the negative emotions that arise from it.

Today, depression, especially youth depression is becoming more common, and it has turned into a global problem. The proportion of adolescents suffering from depression is increasing year by year, and the proportion of severe depression is also increasing [12,46]. In recent years, the Chinese government and some health-related organizations have consciously strengthened the promotion of common sense education on depression, and when conducting mental health censuses, they have focused on the youth groups in school. The youth group is wider, interacts more closely with society, and faces more diverse and more severe emotional pressures. Mental health issues of youth also require adequate attention from the government. Based on the above conclusions, the author puts forward some policy implications of this research. First, the government should guide and create a clear online environment and give netizens an online space for full expression and effective interaction. Netizens need to improve their digital literacy, exercise their ability to identify false information, and create a healthy online culture. Second, the government can directly use the Internet as a medium to popularize the knowledge of depression, improve people’s mental health awareness, break the fantasy or panic about depression, prevent more young people from falling into a state of depression, and create a beneficial network environment for depression patients. Third, take care of the mental health of older young people and reduce the survival pressure of target groups through policy benefits. Fourth, advocate the reduction of gender discrimination, break the inherent association between specific occupations and specific genders, and focus on gender balance in important areas.

## 6. Conclusions

The results of this study show that the use of mobile Internet can effectively reduce the prevalence of depression in Chinese young adults. This effect was significant in people with higher levels of depression, older age, and who use the Internet for communicative purposes. However, there is no significant difference in mobile Internet use between the sexes. In addition, the use of mobile Internet can improve young people’s sense of social comparison, making them less aware of depression. Our findings help to improve the public and government’s understanding of depression in young adults, and provide suggestions for the prevention and relief of depression.

## 7. Limitations

Finally, some shortcomings of this paper are proposed. First, through the instrumental variable method, the results we obtained are only statistically significant at the 0.1 level, and the significance needs to be improved. Second, the R-square of the regression equation is small, indicating that the explanatory power of the independent variable for the dependent variable is weak. Third, the choice of instrumental variables may be questioned. Although both instrumental variables passed the exogeneity test, there may still be potential influencing factors that have not been identified. Fourth, in the analysis of the mediating mechanism, the influence path of lower socioeconomic status self-assessment on the depression level of young people has not been clarified, and whether there are other factors also need to be further theoretical exploration and rigorous empirical analysis. Fifth, the interpretation of the statistical results in this paper only stays at the conjecture step, and the in-depth interview method of qualitative research can be appropriately added to examine in detail the emotional fluctuations of young people in the process of mobile Internet access, and explore the deep mechanism of using mobile Internet to reduce depression in young people. Sixth, this study is retrospective. The physical and mental state of the respondents at that time will affect the authenticity and accuracy of the data. The article scores and analyzes the self-reported depressive symptoms of the respondents. In fact, however, psychiatric history, comorbidities, or factors related to potential pathogenesis can affect depression score. We have not taken these factors into account.

## Figures and Tables

**Figure 1 ijerph-19-04473-f001:**
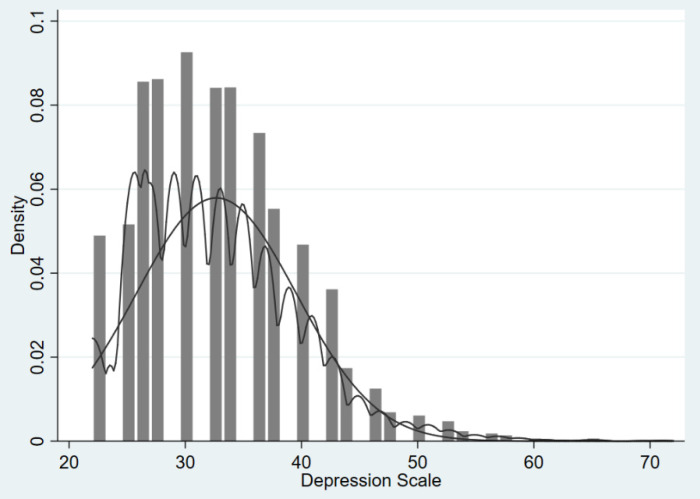
Histogram for depression scale distribution, overlaid with a kernel density and a normal density.

**Figure 2 ijerph-19-04473-f002:**
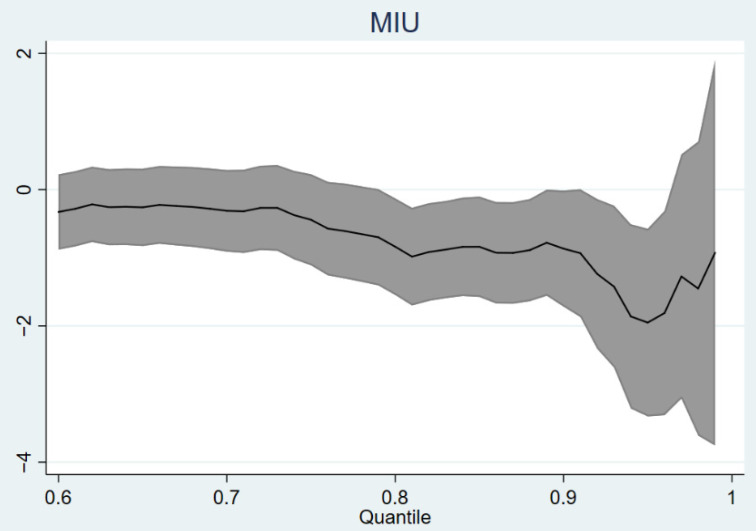
Quantile regression coefficient plot using mobile internet.

**Figure 3 ijerph-19-04473-f003:**
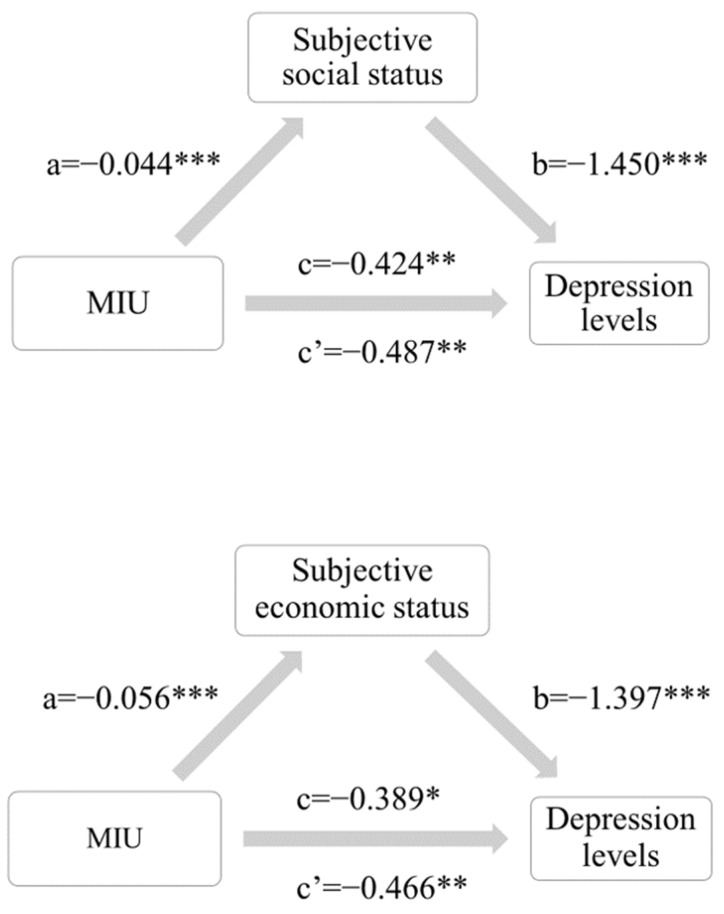
The influence of subjective social status and subjective ecnomic status on depression levels. * *p* < 0.05, ** *p* < 0.01, *** *p* < 0.001.

**Table 1 ijerph-19-04473-t001:** Descriptive statistics.

Variables	Mean	S.D.	N
depression	32.67	6.890	10,499
MIU	0.87	0.337	10,499
age	29.45	6.185	10,499
gender	0.50	0.500	10,499
current marital status	0.68	0.468	10,499
place of residence	0.75	0.432	10,499
employment	0.86	0.346	10,499
perceived wealth gap	0.78	0.414	10,499

**Table 2 ijerph-19-04473-t002:** Correlation matrix of the variables.

	Depression	MIU	Age	Gender	Marital Status	Place of Residence	Employment	Perceived Wealth Gap
depression	1	−0.03 *	0.07 *	−0.04 *	0.01	0.05 *	−0.05 *	0.01
MIU	−0.04 *	1	−0.15 *	0.03 *	−0.09 *	−0.13 *	0.01	0.12 *
age	0.07 *	−0.15 *	1	0.03 *	0.59 *	−0.07 *	0.03 *	−0.02
gender	−0.04 *	0.03 *	0.02 *	1	−0.13 *	−0.00	0.26 *	0.01
marital status	0.01	−0.09 *	0.60 *	−0.13 *	1	0.03 *	−0.12 *	−0.03 *
place of residence	0.05 *	−0.13 *	−0.07 *	−0.00	0.03 *	1	−0.06 *	−0.08 *
employment	−0.05 *	0.01	0.02 *	0.26 *	−0.12 *	−0.06 *	1	0.02 *
perceived wealth gap	0.01	0.12 *	−0.02	0.01	−0.03 *	−0.08 *	0.02 *	1

Note: * *p* < 0.05.

**Table 3 ijerph-19-04473-t003:** OLS regressions.

	Model 1	Model 1
MIU	−0.843 ***	−0.424 **
	(−4.22)	(−2.06)
age		0.002 ***
		(9.15)
gender		−0.503 ***
		(−3.62)
place of residence		0.559 ***
		(3.42)
marital status		−0.879 ***
		(−4.93)
employment		−1.010 ***
		(−5.01)
perceived wealth gap		0.323 **
		(2.00)
provincial dummies	No	Yes
Constant	33.401 ***	32.070 ***
	(179.44)	(45.73)
Observations	10,499	10,499
R^2^	0.002	0.033

Note: ** *p* < 0.01, *** *p* < 0.001.

**Table 4 ijerph-19-04473-t004:** Instrument variable regressions.

	Depression OLS	Depression OLS	MIU 2SLS—First Stage	MIU 2SLS—First Stage	MIU 2SLS—Second Stage
attitude towards the Internet	−0.248		0.185 ***		
	(−1.59)		(25.82)		
household telecommunication charges		−0.127		0.032 ***	
		(−1.41)		(7.63)	
MIU					−1.526 *
					(−1.85)
age	0.002 ***	0.002 ***	−0.000 ***	−0.000 ***	0.002 ***
	(9.25)	(9.41)	(−12.17)	(−14.54)	(7.29)
gender	−0.513 ***	−0.515 ***	0.024 ***	0.023 ***	−0.481 ***
	(−3.69)	(−3.70)	(3.73)	(3.49)	(−3.42)
place of residence	0.563 ***	0.552 ***	−0.060 ***	−0.076 ***	0.452 **
	(3.44)	(3.37)	(−7.95)	(−9.80)	(2.56)
marital status	−0.883 ***	−0.844 ***	0.011	−0.003	−0.878 ***
	(−4.96)	(−4.63)	(1.39)	(−0.33)	(−4.91)
employment	−1.003 ***	−1.014 ***	−0.006	−0.003	−1.023 ***
	(−4.97)	(−5.01)	(−0.67)	(−0.36)	(−5.06)
perceived wealth gap	0.325 **	0.295 *	0.049 ***	0.074 ***	0.404 **
	(2.00)	(1.82)	(6.55)	(9.64)	(2.34)
provincial dummies	Yes	Yes	Yes	Yes	Yes
Constant	31.848 ***	32.298 ***	0.891 ***	0.892 ***	33.257 ***
	(46.71)	(39.67)	(28.39)	(23.24)	(30.27)
Observations	10,499	10,443	10,499	10,443	10,443
R^2^	0.033	0.033	0.142	0.092	0.031

Note: * *p* < 0.05, ** *p* < 0.01, *** *p* < 0.001.

**Table 5 ijerph-19-04473-t005:** Quantile regression 1.

	(1)	(2)	(3)	(4)	(5)
	0.1	0.25	0.5	0.75	0.9
MIU	−0.000	−0.357	−0.218	−0.449	−0.865 **
	(−0.00)	(−1.43)	(−0.83)	(−1.42)	(−2.14)
age	0.000	0.001 ***	0.002 ***	0.003 ***	0.003 ***
	(0.00)	(5.05)	(7.41)	(7.48)	(6.37)
gender	0.000	−0.615 ***	−0.668 ***	−0.697 ***	−0.473 *
	(0.00)	(−3.65)	(−3.75)	(−3.26)	(−1.74)
place of residence	0.000	0.307	0.670 ***	0.579 **	0.734 **
	(0.00)	(1.55)	(3.20)	(2.30)	(2.29)
marital status	−0.000	−0.423 *	−0.779 ***	−1.192 ***	−1.152 ***
	(−0.00)	(−1.96)	(−3.41)	(−4.34)	(−3.30)
employment	−0.000	−0.639 ***	−1.151 ***	−0.853 ***	−1.195 ***
	(−0.00)	(−2.61)	(−4.46)	(−2.75)	(−3.02)
perceived wealth gap	0.000	0.575 ***	0.218	0.275	0.198
	(0.00)	(2.92)	(1.05)	(1.10)	(0.62)
provincial dummies	Yes	Yes	Yes	Yes	Yes
Constant	22.000 ***	25.970 ***	31.680 ***	37.733 ***	41.355 ***
	(26.49)	(30.52)	(35.28)	(34.93)	(30.09)
Observations	10,499	10,499	10,499	10,499	10,499

Note: * *p* < 0.05, ** *p* < 0.01, *** *p* < 0.001.

**Table 6 ijerph-19-04473-t006:** Quantile regression 2.

	(1)	(2)	(3)	(4)	(5)	(6)
	0.7	0.75	0.8	0.85	0.9	0.95
MIU	−0.311	−0.449	−0.837 **	−0.839 **	−0.865 **	−1.949 ***
	(−1.06)	(−1.42)	(−2.48)	(−2.25)	(−2.14)	(−3.32)
age	0.003 ***	0.003 ***	0.003 ***	0.003 ***	0.003 ***	0.003 ***
	(7.91)	(7.48)	(7.80)	(6.83)	(6.37)	(4.88)
gender	−0.651 ***	−0.697 ***	−0.385 *	−0.300	−0.473 *	−0.584
	(−3.28)	(−3.26)	(−1.69)	(−1.20)	(−1.74)	(−1.47)
place of residence	0.651 ***	0.579 **	0.664 **	0.666 **	0.734 **	0.912 *
	(2.79)	(2.30)	(2.47)	(2.25)	(2.29)	(1.96)
marital status	−1.159 ***	−1.192 ***	−1.360 ***	−1.286 ***	−1.152 ***	−1.506 ***
	(−4.56)	(−4.34)	(−4.65)	(−4.00)	(−3.30)	(−2.96)
employment	−0.987 ***	−0.853 ***	−0.934 ***	−1.292 ***	−1.195 ***	−1.981 ***
	(−3.43)	(−2.75)	(−2.82)	(−3.55)	(−3.02)	(−3.45)
perceived wealth gap	0.163	0.275	0.267	0.306	0.198	0.453
	(0.71)	(1.10)	(1.00)	(1.05)	(0.62)	(0.98)
provincial dummies	Yes	Yes	Yes	Yes	Yes	Yes
Constant	36.096 ***	37.733 ***	39.419 ***	40.326 ***	41.355 ***	44.153 ***
	(36.08)	(34.93)	(34.24)	(31.83)	(30.09)	(22.09)
Observations	10,499	10,499	10,499	10,499	10,499	10,499

Note: * *p* < 0.05, ** *p* < 0.01, *** *p* < 0.001.

**Table 7 ijerph-19-04473-t007:** Heterogeneous effects 1.

	(1)	(2)
	18–28 years old	29–40 years old
MIU	−1.667	−1.738 *
	(−0.94)	(−1.85)
Constant	33.160 ***	35.862 ***
	(16.35)	(24.01)
Observations	4524	5919
R^2^	0.022	0.044

Note: * *p* < 0.05, *** *p* < 0.001.

**Table 8 ijerph-19-04473-t008:** Heterogeneous effects 2.

	Female	Male
MIU	−1.320	−1.555
	(−1.24)	(−1.21)
Constant	31.937 ***	34.334 ***
	(21.57)	(20.56)
Observations	5264	5179
R^2^	0.036	0.035

Note: *** *p* < 0.001.

**Table 9 ijerph-19-04473-t009:** Heterogeneous effects 3.

Purpose of Use	Work	Social Interaction	Entertainment	Business Activities
depression	−1.358	−3.854 **	−3.347 *	−1.360
	(−1.58)	(−2.01)	(−1.93)	(−1.55)
Constant	32.664 ***	35.134 ***	34.777 ***	32.518 ***
	(27.21)	(18.38)	(19.23)	(35.15)
Observations	7005	9222	9223	9223
R^2^	0.023	0.013	0.002	0.018

Note: * *p* < 0.05, ** *p* < 0.01, *** *p* < 0.001.

**Table 10 ijerph-19-04473-t010:** Benchmark regression results (16–40 years old).

	Model 1	Model 2
MIU	−0.767 ***	−0.379 *
	(−3.99)	(−1.92)
age		0.002 ***
		(9.59)
gender		−0.525 ***
		(−3.92)
place of residence		0.586 ***
		(3.68)
marital status		−0.883 ***
		(−5.01)
employment		−1.085 ***
		(−5.51)
perceived wealth gap		0.319 **
		(2.04)
provincial dummies	No	Yes
Constant	33.295 ***	31.929 ***
	(185.83)	(46.95)
Observations	11,177	11,177
R^2^	0.001	0.033

Note: * *p* < 0.05, ** *p* < 0.01, *** *p* < 0.001.

**Table 11 ijerph-19-04473-t011:** Instrumental variable regression results (16–40 years).

	(1)	(2)	(3)	(4)	(5)
	depression OLS	depression OLS	MIU 2SLS—First stage	MIU 2SLS—First stage	MIU 2SLS—Second stage
attitude towards the Internet	−0.226		0.182 ***		
	(−1.52)		(26.25)		
household telecommunication charges		−0.102		0.032 ***	
		(−1.17)		(7.64)	
MIU					−1.413 *
					(−1.76)
age	0.002 ***	0.002 ***	−0.000 ***	−0.000 ***	0.002 ***
	(9.69)	(9.79)	(−11.17)	(−12.97)	(7.98)
gender	−0.533 ***	−0.535 ***	0.024 ***	0.024 ***	−0.501 ***
	(−3.99)	(−3.98)	(3.91)	(3.74)	(−3.70)
place of residence	0.587 ***	0.580 ***	−0.057 ***	−0.073 ***	0.488 ***
	(3.69)	(3.63)	(−7.68)	(−9.55)	(2.86)
marital status	−0.886 ***	−0.859 ***	0.014 *	0.002	−0.878 ***
	(−5.03)	(−4.77)	(1.73)	(0.23)	(−4.96)
employment	−1.080 ***	−1.089 ***	−0.005	−0.004	−1.097 ***
	(−5.48)	(−5.51)	(−0.57)	(−0.40)	(−5.56)
perceived wealth gap	0.321 **	0.293 *	0.050 ***	0.074 ***	0.394 **
	(2.05)	(1.88)	(6.80)	(9.94)	(2.36)
provincial dummies	Yes	Yes	Yes	Yes	Yes
Constant	31.737 ***	32.084 ***	0.876 ***	0.872 ***	33.020 ***
	(47.96)	(40.63)	(28.44)	(23.19)	(31.30)
Observations	11,177	11,116	11,177	11,116	11,116
R^2^	0.033	0.034	0.134	0.085	0.031

Note: * *p* < 0.05, ** *p* < 0.01, *** *p* < 0.001.

**Table 12 ijerph-19-04473-t012:** Benchmark instrumental variable regression results.

	(1)	(2)	(3)	(4)	(5)
	depression OLS	depression OLS	MIU 2SLS—First stage	MIU 2SLS—First stage	MIU 2SLS—Second stage
use of stationary computer	−1.185 ***	−0.679 ***			−1.448 *
	(−8.77)	(−4.56)			(−1.90)
attitude towards the Internet			0.197 ***		
			(19.65)		
household telecommunication charges				0.042 ***	
				(7.05)	
age		0.002 ***	−0.000 ***	−0.000 ***	0.002 ***
		(8.86)	(−11.56)	(−13.48)	(6.74)
gender		−0.472 ***	0.061 ***	0.060 ***	−0.428 ***
		(−3.39)	(6.80)	(6.60)	(−2.92)
place of residence		0.413 **	−0.242 ***	−0.259 ***	0.191
		(2.47)	(−22.95)	(−24.07)	(0.74)
marital status		−0.952 ***	−0.101 ***	−0.117 ***	−1.040 ***
		(−5.33)	(−8.80)	(−9.82)	(−5.34)
employment		−0.898 ***	0.159 ***	0.161 ***	−0.784 ***
		(−4.42)	(12.23)	(12.13)	(−3.31)
perceived wealth gap		0.361 **	0.075 ***	0.100 ***	0.437 **
		(2.23)	(7.15)	(9.41)	(2.44)
provincial dummies	No	Yes	Yes	Yes	Yes
Constant	33.184 ***	32.151 ***	0.602 ***	0.570 ***	32.776 ***
	(371.84)	(47.56)	(13.70)	(10.69)	(36.64)
Observations	10,499	10,499	10,499	10,443	10,443
R^2^	0.007	0.035	0.225	0.199	0.033

Note: * *p* < 0.05, ** *p* < 0.01, *** *p* < 0.001.

**Table 13 ijerph-19-04473-t013:** Statistics for the intermediates considered.

	Full Sample	User	Non-User	Diff.
subjective well-being	0.813	0.830	0.703	−0.127 ***
Observations	10,497	9129	1368	
interpersonal self-assessment	0.764	0.777	0.674	−0.103 ***
Observations	10,498	9130	1368	
Subjective social-status	0.187	0.177	0.251	0.074 ***
Observations	10,485	9121	1364	
subjective economic status	0.178	0.168	0.239	0.071 ***
Observations	9566	8257	1309	

Note: *** *p* < 0.001.

**Table 14 ijerph-19-04473-t014:** Mediation effect 1.

	Subjective Well-Being	Interpersonal Self-Assessment	Subjective Social Status	Subjective Economic Status
MIU	0.093 ***	0.063 ***	−0.044 ***	−0.056 ***
	(8.05)	(4.95)	(−3.75)	(−4.72)
Constant	0.748 ***	0.643 ***	0.190 ***	0.211 ***
	(18.98)	(14.93)	(4.76)	(5.28)
Observations	10,497	10,498	10,485	9566
R^2^	0.046	0.041	0.025	0.021

Note: *** *p* < 0.001.

**Table 15 ijerph-19-04473-t015:** Mediation effect 2.

	(1)	(2)	(3)
	depression	depression	depression
MIU	−1.413 *	−0.4873 **	−0.466 **
	(−1.76)	(0.21)	(0.21)
subjective social-status subjective economic status		−1.450 ***	−1.397 ***
		(0.17)	(0.18)
Constant	33.257 ***	37.327 ***	38.084 ***
	(30.27)	(2.79)	(2.61)
Observations	10,443	10,485	9566

Note: * *p* < 0.05, ** *p* < 0.01, *** *p* < 0.001.

## Data Availability

This paper uses data from the Chinese Family Panel Studies (CFPS), a nationally representative longitudinal survey funded by Peking University and the National Natural Science Foundation of China and initiated by the Institute of Social Sciences of Peking University.

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
