# Peer review of "Does Mobile Internet Use Affect the Depression of Young Chinese Adults? An Instrumental Variable Quantile Analysis"

_ijerph, 2022, doi:10.3390/ijerph19084473_

Round 1
Reviewer 1 Report
The topic of this research is interesting and up-to-date. Therefore, I recommend the authors to deeply revise the article according to my observations in the review.

Author Response
Thank you very much for your valuable comments. 1. Abstract: Lines 16-18: Authors mention the mechanism of social comparison. However, there is no measurement of this process in the manuscript. Is it possible to rephrase this conclusion? The paper uses subjective social status and subjective economic status as the indicators of social comparison. See lines 305-311 for the measurement of these two variables. (More detailed discussion below) 2. Introduction: All provided data should be accompanied by citations Lines 23-24:earliest onset of all types of depression is around the age of 14. The prevalence is 3 times higher in individuals aged 18-29 than those aged 60 and older. References have been added to the text. Cited from: Huang YQ, 2021, Studies on the burden of mental disorder disorders and health service utilization in China, China Science And Technology Achievements, 2:75-76. Lines 29-31: Compared with the data of previous years, all indicators have maintained steady growth, indicating that the mobile Internet has become an indispensable and important tool in modern life. References have been added to the text. Cited from: China Internet Network Information Center, The 47th Statistical Report on China's Internet Development, and previous reports in the same series. Check the entire manuscript for missing citations. Also, citations should be adjusted (see the journal instructions for authors it is not APA format). The appropriate citation and reference style should be used. The format of references has been modified. Lines 37-39: Authors should provide a reference for this affirmation and explain in more details the association/ correlation between internet use and psychotic disorders, which are severe mental problems involving complex etiopathological mechanisms/ risk factors. For a long time, people have both positive and negative views on Internet use. Opponents believe that increasing proportion of young adults with psychotic disorders is related to inappropriate use of the Internet. Add between line 38-39: Improper Internet use will lead to a variety of pathological characteristics such as increased dependence, social withdrawal, guilt and difficulty in inhibiting Internet desire, and further lead to depressive disorder, anxiety disorder as well as depression and anxiety disorder (comorbid disorder). Luo et al. (2005) tested 75 adolescents who met the DSM-IV standards. They found that most of the Internet addicted adolescents are comorbidities of depression and anxiety, requiring both anti-depressants and anti-anxiety drugs for psychotherapy. Add references: Luo Kang-ling, MengHua-qing, Fu Yi-xiao, 2005, Depression and anxiety as well as the comorbid of them inadolescents with internet addiction disorder, Chinese Journal of Tissue Engineering Research, 8:4-5. Lines 53-54: Is it the incidence, or rather the prevalence of depression? Replaced with: the prevalence of depression. Line 66: I recommend the authors to replace the expression "controlling the risk of depression” with a more scientifically accurate statement. Replaced with: controlling the risk →the prevalence of depression. Lines 63-66: The objective of this study is not clear. Please try to reformulate this part. Lines 63-70 have been revised to the following: This article attempts to show the mental health status of young people through statistical data, exploring how the Internet and its different purposes of use have emotional effects, and provide a reference for actually solving the young adults’ depression and controlling the risk of depression. This paper attempts to use an instrumental variable approach to explore the causal association and the underlying mechanisms between the mobile Internet use (MIU) and depression in young adults(aged 18-40 years), contributing empirical experience to the current field and providing possible policy recommendations. On the one hand, this paper focuses on the mental health problems of young people, especially the specific topic of depression, and contributes empirical experience to the current field. On the other hand, in view of the increasingly serious situation of depression in China, we attempt to provide possible policy recommendations in the field of mental health, to provide a reference for actually solving the young adults ’depression and reduce the prevalence of depression. 3. Prior literature: Lines 72-75: Could authors provide a more comprehensive description of depression, as stated by international systems for classification of mental disorders (e.g., DSM-5 or ICD-11)? add between line 74-75: ICD-11 and DSM-5 have basically the same diagnostic description of depressive episode, including depressive mood, loss of interest, accompanied by other cognitive, behavioral, or autonomic symptoms, which has a significant impact on the level of individual social function. Add References: American Psychiatric Association, 2015, DSM-5, Lines 75-82: In my opinion, a more detailed description of the social impact of depression in young adults would be necessary. add between line 79-80: Depressive symptoms can negatively affect a person's mood, thinking, self feeling, interpersonal communication and physical function (Dozois&Dobson,2004), leading to social isolation and even an increase in the level of suicidal ideation (Kim et al. 2006). At present, the number of common mental disorders and psychological problems in China is increasing year by year, and malignant cases caused by personal extreme emotions occur from time to time. It is harmful to social stability, interpersonal harmony, and public happiness. Line 101: Please define the "online big data platform". In the cited document, the big data platforms used by the author are baidu.com and Sina Microblog, which are the most popular search engines and social platforms in China, respectively. The author calculated the daily search volume of depression through Baidu Index (BDI) and Sina Microblog index (SMI). 4. Data and Methods: Please offer more details about the sampling method, as well as the data curation procedure. In 2008 and 2009, CFPS conducted initial and follow-up surveys in Beijing, Shanghai and Guangdong, then officially conducted investigations in 2010. All baseline family members defined by the 2010 baseline survey and their future consanguineous / adopted children will be regarded as genetic members of CFPS and become permanent tracking objects. The fifth round of national survey (cfps2018) began on June 5, 2018 and lasted until May 2019. The sample covers 25 provinces / cities / autonomous regions in China, including about 95% of the population. It adopts implicit stratification, multi-stage, multi-level probability proportional to size (PPS). Therefore, the sample of CFPS can be regarded as a national representative sample. There is no information about the ethical aspects of this research, namely informed consent of included participants and an official ethical approval of the study (see the journal requirements for the existence of an Informed Consent Statement, as well as Research and Publication Ethics). It is implemented by the Institute of Social Science Survey (iSSS) of Peking University, and funded by Peking University and the National Natural Science Foundation of China. The confidentiality regulations are stated at the beginning of the questionnaire: according to Article 25 of Chapter III of the statistics law, "the data obtained in the statistical survey that can identify or infer the identity of a single statistical survey object shall not be provided or disclosed by any unit or individual, and shall not be used for purposes other than statistics". Lines 126-132: A more detailed description of the used questionnaires it is strongly recommended, including type of instruments, psychometric properties (validity, reliability, internal construct, etc), and few item examples. Also, the rating of responses and total score calculation are not clear. Please be more specific regarding these aspects. In CFPS2018, the Center for Epidemiologic Studies Depression Scale (CES-D) was used to test the individual's depression level, and all of them were converted to the simplified 8-question version (CES-D8). l felt depressed/l felt that everything I did was an effort/My sleep was restless/l was happy/l felt lonely/l enjoyed life/I felt sad/l could not get "going." Please tell me how often you have felt this way during the past week: Rarely or None of the Time (Less than 1 Day) Some or a Little of the Time (1-2 Days) Occasionally or a Moderate Amount of Time (3-4 Days) Most or All of the Time (5-7 Days) The database is publicly published on the network and can be obtained through account registration. The sampling method, type of instruments, scoring scale, data cleaning details and ethical aspects of the questionnaire are displayed on the official website of CFPS(http://www.isss.pku.edu.cn/cfps/index.htm). See also the attachment. Histogram 1 should be more explicit regarding the studied variables Changed to: Figure 1. Histogram for depression scale distribution, overlaid with a kernel density and a normal density At the first occurrence of an abbreviation, it should be explained (for example, Line 170: IVQR。 The full name has been added where IVQR first appeared (line 170): instrumental variables quantile regression. 5. Results: At line 134, authors define MIU as the core independent variable. However, at line 185 and 228, the plural is used. Please clarify the way variables were classified. The plural form has been changed to the singular form: Line 185: First, using instrumental variable regression solves the endogeneity problem of independent variables…… Line 228: have significant effects on the core independent variables. Lines 206-210: This part involves contradictory information. It is reported that MIU has a negative impact on young peoples’ depression levels. However, it is also stated that MIU and depression have a negative correlation, meaning an inverse relationship. The previous sentence means that MIU has a negative impact on depression scores in young people. The higher the depression score, the more severe the depressive symptoms. Changed to: The results suggest that even after controlling for covariates, the negative positive effect of MIU on depression in young adults remained significant. Specifically, the depression of young people who use mobile internet is lower than those of young people who do not use mobile internet. Lines 259-270: It is recommended that authors integrate this part at the Discussions section, and not the Results, since it highlights previous research that is in line with the outcomes of this article. Move lines 259-263 between lines 395 and 396: Using data from the nationally representative surveys of U.S. adolescents in grades 8 through 12 (N = 506,820), Twenge et al. (2018) document that the use of the Internet is more likely to increase depressive symptoms only among girls, because the Internet may have larger effects on adolescent girls’ mood than on boys’. Mars et al. (2020) also find that high levels of internet use is more likely to increase females’ risk of depression. Lines 263-270 are slightly modified and remain in the original position: However, in China , the effect of MIU on depression in male and female was not statistically significant, meaning that there was no gender difference(see Table 8). Through the two-sample t-test, we found that there are differences in the preference of the two genders for using the Internet. Female are more likely to engage in online work and online business activities, and male are more inclined to use the Internet for entertainment. There is no significant difference in online social behavior between male and female. 6. Conclusions and Discussions I suggest that authors address the Discussions and Conclusions of the article as separated structures, according to the journal requirements on Manuscript Sections. Conclusions should be formulated as an individual section at the end, after the limitations. The idea of this paper is to put forward a conclusion, make a series of conjectures and discussions, and then elaborate the next conclusion. If the conclusion is separated from the discussion part, it may cause repeated conclusions. Please offer some citations to support your statements (for example, see Lines 377-379: “On the contrary, relatively more depressed groups tend to self-isolate in real life and stay in a lonely environment for a long time, so they are more dependent on the Internet and pay more attention to the functionality of the Internet"). The entire Discussions section should be checked for missing citations. (Young & Rogers, 1998; Ko et al., 2009; Xun 2013) Ko CH, Yen JY, Chen CS, et al. Predictive values of psychiatric symptoms for internet addiction in adolescents: A 2-year prospective study. Archives of Pediatrics and Adolescent Medicine, 2009, 163(10): 937-943 Line 409: Is the notion of netizens integrated as a description of young adults using Internet? Netizens (Internet citizens) generally refer to all people who conduct online activities through computers and the Internet, including young adults. The concept of social comparisons is mentioned at Introduction and Discussions. However, it is not addressed at Results, specifically Mechanisms part. Add between lines 437 and 438: We believe that the decline of subjective social status and subjective economic status is related to social comparison. People could obtain social networking profiles presented by others on Internet platforms. Compared with their own situation, the inevitable comparisons erode their self-esteem (Zhou, 2017). The outcomes of the present study should be more clearly elaborated and supported/ linked to previous research. Add between line 365 and 366: Previous studies on the impact of MIU on depression are mostly concentrated in the elderly and adolescents, and the attention to youth is relatively lacking. At the same time, the existing academic achievements have not focused on the distribution of the impact of MIU on young people, and have not been able to solve the endogenous problem. Line 451: Is there a single author? Modified to: In addition, the author believes we believe that with the popularity of the Internet Please check the expression in English throughout the entire manuscript (for example, see Lines 460-461: "Today, depression is becoming younger and more common, and it has become a global problem”). Modified to: Today, depression, especially youth depression is becoming younger and more common, and it has become turn into a global problem. From my opinion, the major limitation of this study is that it was retrospective and did not include assessments of other aspects that could influence depression scores, like psychiatric history, comorbidities or factors related to potential onset mechanism. I propose that authors also discuss this limitation. Add between lines 494-496: Sixth, this study is retrospective. The physical and mental state of the respondents at that time will affect the authenticity and accuracy of the data. The article scores and analyzes the self-reported depressive symptoms of the respondents. In fact, however, psychiatric history, comorbidities or factors related to potential pathogenesis can affect depression score. We have not taken these factors into account. Authors contribution is incomplete (for example, the writing/ original draft preparation contribution is not stated). Yuyan Chen: conceptualisation, methodology, software, formal analysis, data curation, writing—original draft preparation, and visualisation. Lin Wu: validation, investigation, resources, and writing—review and editing. Zenghua Guo: conceptualization, methodology, writing—original draft preparation, supervision, project administration and funding acquisition of the study. All authors have read and agreed to the published version of the manuscript. Statements about funding, acknowledgement, data availability and conflict of interest are missing. Conflict of Interest: The authors declare that the research was conducted in the absence of any commercial or financial relationships that could be construed as a potential conflict of interest. 7. References References should be checked. Overall, there are 71 references, but only 27 are cited in the manuscript. Also, there are at least 6 citations in the text body that are not included at References. References have been added or deleted. The citation style should follow the journal instructions for authors. The citation style has been modified.
Reviewer 2 Report
39 - 45: The sentence is awkward, lengthy, and needs rewording for clarity.
72-74: Authors should clarify that the listed terms are "defining characteristics" of depression, not the definition of the disorder.
81: Missing "adults"
85, 90: Nouns should be plural
Spacing issues present in the paper (Example: lines 78, 82, 351) can be distracting
390-394, 414-417 - Provide support for perceived masculinity and femininity of certain online activities
476: "Network atmosphere" is used as a full sentence, but seems out of place. Reword or remove.
Author Response
Thank you very much for your valuable comments.
39 - 45: The sentence is awkward, lengthy, and needs rewording for clarity.
changed to:
Internet is free from the constraints of time and space and has the advantages of maintaining low cost of cross-border communication. therefore, netizens shift from intuitive, personality-dependent face-to-face communication to fragmented, text-based visual platform communication. The weak relationship with strangers has impacted their strong relationship with close people, often resulting in a withered communication situation, misunderstandings, negative emotions, and other adverse consequences.
72-74: Authors should clarify that the listed terms are "defining characteristics" of depression, not the definition of the disorder.
Depression is referred to as perceived pessimistic mood (Tan et al., 2015) or academically defined as a feeling of helplessness and hopelessness (Scheff, 2001), the level of feelings of guilt and worthlessness (Nolen-Hoeksema, 2000), or psychomotor retardation (Chaudhari et al., 2015).
81: Missing "adults"
added "adults"
85, 90: Nouns should be plural
adolescent→adolescents
young adult→young adults
Spacing issues present in the paper (Example: lines 78, 82, 351) can be distracting
Spacing issues has been adjusted.
390-394, 414-417 - Provide support for perceived masculinity and femininity of certain online activities
In recent years, female players have gradually risen and occupied nearly half of the video game market. Video games are no longer exclusive to men. The 2019 Research Report on China's mobile game industry shows that the proportion of men and women in China's mobile game users has stabilized as a whole, and the proportion of women users has generally remained at about 42% [96-1]. In 2020, a statistics of the United States also showed that female players accounted for 41% of the total number of game users in the United States [96-2]. Shen et al. (2016) Proved that there is no difference in game ability between female and male players, and women's progress in learning game skills is at least as fast as men [96-6]. At the same time, a study on body mimicry behavior in cyberspace has proved that men show a neutral tendency when fabricating virtual game images, that is, they choose women as their virtual images [95]
Yi et al. (2022) investigated the digital literacy level of 335 full-time teachers in higher vocational colleges in Zhejiang Province through random sampling. The results show that the digital literacy of female teachers is slightly higher than that of male teachers, but this difference does not reach a significant level [98]. Zhu et al. (2019) conducted a questionnaire survey with humanities scholars as the research object, and found that except for the primary basic information ability, there was no significant difference in the intermediate and advanced digital ability of scholars of different genders.
Although there are few empirical studies on the topic of digital literacy in China, we consider that the difference between the sexes in Internet use has gradually shrunk, and this trend is quite obvious.
476: "Network atmosphere" is used as a full sentence, but seems out of place. Reword or remove.
Changed 474-475 to:……and also create a beneficial network environment for depression patients. Network atmosphere.

Reviewer 3 Report
It is a very interesting and well written paper
Comments for the authors
INTRODUCTION SECTION
- “Opponents believe…. inappropriate use of internet” : please add a reference accordingly
- “Existing academic….due to internet use in China are relatively rare”: it is suggested this sentence to be rewritten, so as to give a clear meaning
SECTION PRIOR LITERATURE
it is recommended that authors
- add a small paragraph referring to the functional impairment of adolescents and young adults due to depression
- expand the paragraph regarding the association between internet and depression (positive and negative)
- ''... defined as a feelings..."delete "a"
DATA AND METHODS
- Chinese Family Panel Studies: it s suggested authors to give more information about CFPS and analyze the protocol eg if they used specific measures, if ithe interview was online, the duration etc. Additionally it is recommended authors give information about Ethics Committe approval
- “18-40 years old" it is a very diverse age range. could authors clarify why they selected this range?
CONCLUSIONS AND DISCUSSION
- “This study explores how MUI affects the depression … at different quantiles of depression performance : please replace depression performance with depression symptomatology
- “reduce the degree of depression”: it is supposed authors mean that depressed people feel relief, please rephrase accordingly
- “MIU is associated with an decreased depression”: please replace "decreased depressive feelings"
- “The gender difference in the influence of MIU is not significant, which could be due to the fact that the parenting style of many one-child families in China has changed the traditional gender role expectations. Many females are not only good at interpersonal communication, but also keen on Internet surfing and online games, showing masculinity. And some males also show more femininity.”: please, explain more precisely issues about femininiy and masculinity. Is it authors' state ? if so, please refer that; otherwise add a literature reference
- “ And some men showed more female preference” : please clarify what "the female preference" means
- “Today depression is becoming younger…..severe depression is also increasing”: please refer accordingly
- “Network atmosphere”: please reference accordingly
LIMITATIONS
It is recommended to add a limitation paragraph
Author Response
Thank you very much for your valuable comments.
INTRODUCTION SECTION
“Opponents believe…. inappropriate use of internet” : please add a reference accordingly
add between line 38-39:
Improper Internet use will lead to a variety of pathological characteristics such as increased dependence, social withdrawal, guilt and difficulty in inhibiting Internet desire, and further lead to depressive disorder, anxiety disorder as well as depression and anxiety disorder (comorbid disorder). Luo et al. (2005) tested 75 adolescents who met the DSM-IV standards. They found that most of the Internet addicted adolescents are comorbidities of depression and anxiety, requiring both antidepressants and antianxiety drugs for psychotherapy.
Add references:
Luo Kang-ling, MengHua-qing, Fu Yi-xiao, 2005, Depression and anxiety as well as the comorbid of them inadolescents with internet addiction disorder, Chinese Journal of Tissue Engineering Research, 8:4-5.
61-63“Existing academic….due to internet use in China are relatively rare”: it is suggested this sentence to be rewritten, so as to give a clear meaning
Modified to:
The existing academic achievements on the impact of MIU on depression has achieved complex results, but most scholars focus on the elderly and adolescents, rather than the youth.
SECTION PRIOR LITERATURE
it is recommended that authors
- add a small paragraph referring to the functional impairment of adolescents and young adults due to depression
add between line 79-80:
Depressive symptoms can negatively affect a person's mood, thinking, self feeling, interpersonal communication and physical function (Dozois&Dobson,2004), leading to social isolation and even an increase in the level of suicidal ideation (Kim et al. 2006).
At present, the number of common mental disorders and psychological problems in China is increasing year by year, and malignant cases caused by personal extreme emotions occur from time to time. It is harmful to social stability, interpersonal harmony, and public happiness.
- ''... defined as a feelings..."delete "a"
Deleted "a"
DATA AND METHODS
Chinese Family Panel Studies: it’s suggested authors to give more information about CFPS and analyze the protocol eg if they used specific measures, if ithe interview was online, the duration etc. Additionally it is recommended authors give information about Ethics Committee approval
The fifth round of national survey (cfps2018) began on June 5, 2018 and lasted until May 2019. The sample covers 25 provinces / cities / autonomous regions in China, including about 95% of the population. It adopts implicit stratification, multi-stage, multi-level probability proportional to size (PPS). Therefore, the sample of CFPS can be regarded as a national representative sample.
The baseline survey of CFPS in 2010 was conducted in the form of face-to-face interview, and CAPI (Computer Assisted Personal Interviewing) was used. Since the follow-up survey in 2012, the mixed survey mode dominated by face-to-face interview and supplemented by telephone interview has been implemented, and the CATI (Computer Assisted Telephone Interview) interview mode has been added.
In CFPS2018, the Center for Epidemiologic Studies Depression Scale (CES-D) was used to test the individual's depression level, and all of them were converted to the simplified 8-question version (CES-D8).
l felt depressed/l felt that everything I did was an effort/My sleep was restless/l was happy/l felt lonely/l enjoyed life/I felt sad/l could not get "going."
Please tell me how often you have felt this way during the past week:
Rarely or None of the Time (Less than 1 Day)
Some or a Little of the Time (1-2 Days)
Occasionally or a Moderate Amount of Time (3-4 Days)
Most or All of the Time (5-7 Days)
It is implemented by the Institute of Social Science Survey (iSSS) of Peking University, and funded by Peking University and the National Natural Science Foundation of China.
The confidentiality regulations are stated at the beginning of the questionnaire: according to Article 25 of Chapter III of the statistics law, "the data obtained in the statistical survey that can identify or infer the identity of a single statistical survey object shall not be provided or disclosed by any unit or individual, and shall not be used for purposes other than statistics".
The database is publicly published on the network and can be obtained through account registration. The sampling method, type of instruments, scoring scale, data cleaning details and ethical aspects of the questionnaire are displayed on the official website of CFPS(http://www.isss.pku.edu.cn/cfps/index.htm). See also the attachment.
“18-40 years old" it is a very diverse age range. could authors clarify why they selected this range? "
We refer to several quantitative research articles from authoritative journals. These articles define the age of youth adults more consistently, with 15, 16 or 18 as the minimum and 35, 39 or 40 as the maximum. It can be considered that in the field of academic research, the age group of 18-40 can better represent young Chinese adults.
CONCLUSIONS AND DISCUSSION
368“This study explores how MIU affects the depression … at different quantiles of depression performance : please replace depression performance with depression symptomatology
Replaced: depression performance → depression symptomatology
380-381“reduce the degree of depression”: it is supposed authors mean that depressed people feel relief, please rephrase accordingly
changed to: Both the use of computer and mobile devices to surf the Internet can significantly reduce the degree of depression among young people make young people less depressed and feel relief.
385“MIU is associated with an decreased depression”: please replace "decreased depressive feelings"
changed to: “MIU is associated with an decreased depression depressive feelings"
390-394“The gender difference in the influence of MIU is not significant, which could be due to the fact that the parenting style of many one-child families in China has changed the traditional gender role expectations. Many females are not only good at interpersonal communication, but also keen on Internet surfing and online games, showing masculinity. And some males also show more femininity.”: please, explain more precisely issues about femininity and masculinity. Is it authors' state ? if so, please refer that; otherwise add a literature reference
The masculinity and femininity mentioned here refer to a traditional rigid gender boundary. Men are generally supposed to be outgoing, independent, and aggressive. Women are supposed to be gentle, sensitive, and considerate. Therefore, video games are generally considered to be the major competition arena of men, while women are considered to be good at expressing their emotions through online chat.
In recent years, female players have gradually risen and occupied nearly half of the video game market. Video games are no longer exclusive to men. The 2019 Research Report on China's mobile game industry shows that the proportion of men and women in China's mobile game users has stabilized as a whole, and the proportion of women users has generally remained at about 42% [96-1]. In 2020, a statistics of the United States also showed that female players accounted for 41% of the total number of game users in the United States [96-2]. Shen et al. (2016) Proved that there is no difference in game ability between female and male players, and women's progress in learning game skills is at least as fast as men [96-6]. At the same time, a study on body mimicry behavior in cyberspace has proved that men show a neutral tendency when fabricating virtual game images, that is, they choose women as their virtual images [95]
Yi et al. (2022) investigated the digital literacy level of 335 full-time teachers in higher vocational colleges in Zhejiang Province through random sampling. The results show that the digital literacy of female teachers is slightly higher than that of male teachers, but this difference does not reach a significant level [98]. Zhu et al. (2019) conducted a questionnaire survey with humanities scholars as the research object, and found that except for the primary basic information ability, there was no significant difference in the intermediate and advanced digital ability of scholars of different genders.
Although there are few empirical studies on the topic of digital literacy in China, we consider that the difference between the sexes in Internet use has gradually shrunk, and this trend is quite obvious [97].
“ And some men showed more female preference”: please clarify what "the female preference" means
What we want to express is that some men go beyond the rigid gender roles and tend to be neutral. Traditionally, feminization of men and masculinization of women are both stigmas. However, male groups with female preferences do not refuse feminization[95], and are even willing to express their femininity. For example, in the process of using the Internet, men can be gentle and considerate, refer costume games rather than e-sports, etc.
If "femininity" can be more appropriately expressed here, we hope to correct "female preference" to "femininity".
459“Today depression is becoming younger…...severe depression is also increasing”: please refer accordingly
The source has been marked in the text:
(Xu et al., 2022) (Guo, 2018)
The following contents are mainly referred to:
Studies have shown that the detection rate of depressive symptoms among middle school students in China is 28.4%, which is higher than the survey results of teenagers in developed countries in Europe, North America and Asia(Xu et al., 2022).
Data from nationwide surveys of the epidemiology of adolescents show that the peak incidence of depression in adolescence is already at the age of 13 to 15 years. Mental health problems among adolescents increased significantly since 1992 to 2014 in China (Guo, 2018).
“Network atmosphere”: please reference accordingly
Changed 474-475 to: ……and also create a beneficial network environment for depression patients. Network atmosphere.
LIMITATIONS
It is recommended to add a limitation paragraph
added a limitation paragraph

Round 2
Reviewer 1 Report
I appreciate authors' cooperation. In my opinion, the current form of the manuscript is significantly improved. However, I still have some remarks regarding this research. Please check my comments in the attachment below.

Author Response
Thank you very much for your valuable comments.
- Abstract:
“It is recommended that authors structure the information in the abstract, including the following: Background, Methods, Results, Discussions, Conclusion.”
changed to:
Background: With the advancement of the digital age, the links between mobile Internet use (MIU) and mental health have attracted the attention of scholars. This paper focuses on the relationship between MIU and depression across the entire distribution of young adults’ depression.
Methods: Based on nationally representative data from the 2018 China Family Panel Studies (CFPS), we explore whether and to what extent MIU affects depression of young adults in China through instrumental variables. In addition, we employ quantile regression approach to explore the depression-MIU gradients and examine potential mediation mechanisms by exploiting variation in several potential intermediates available.
Results: 2SLS estimate suggests that MIU is associated with a decrease in young adults’ depression by 1.526 points. Besides, the effect of MIU was only significantly negative at the 0.8 to 0.96 quantiles.
Discussions: MIU reduces the level of depression in people with higher levels of depression, older age, and who use Internet for communicative purpose. However, there is no significant gender difference in MIU. In addition, young people will improve their feeling of social comparison when using mobile Internet, thus to make them less depressed.
Conclusions: MIU has a significant positive impact on the depression among young Chinese adults.
- Data and Methods:
Please include your answers to our comments regarding the data curation and sampling method in the manuscript.
added:
In 2008 and 2009, CFPS conducted initial and follow-up surveys in Beijing, Shanghai and Guangdong, then officially conducted investigations in 2010. All baseline family members defined by the 2010 baseline survey and their future consanguineous / adopted children will be regarded as genetic members of CFPS and become permanent tracking objects.
The fifth round of national survey (cfps2018) began on June 5, 2018 and lasted until May 2019. The sample covers 25 provinces / cities / autonomous regions in China, including about 95% of the population. It adopts implicit stratification, multi-stage, multi-level probability proportional to size (PPS). Therefore, the sample of CFPS can be regarded as a national representative sample.
CFPS is implemented by the Institute of Social Science Survey (ISSS) of Peking University, and funded by Peking University and the National Natural Science Foundation of China.
Since 2010, CFPS has conducted five rounds of full sample follow-up surveys in 2012, 2014, 2016, 2018 and 2020, which is an authoritative survey and research. The database is publicly published on the network and can be obtained through account registration. However, we haven't got more information about ethical approval yet.
Many important studies have used CFPS data, such as:
- Xu, Q.; Li, J.X.; Yu, X.J. Continuity and Change in Chinese Marriage and the Family Evidence from the CFPS. Chinese Sociological Review 2014, 47, 30-56.
- Zhu, Y.R.; Zhang, S.H.; Yang, J.X. The Effect of Internet Usage on Chinese Residents' Health Behaviours: Evidence from CFPS. Basic & Clinical Pharmacology & Toxicology 2018, 123, 65-65
- Peng, L.; Wang, X.;Ying, S. The Heterogeneity of Beauty Premium in China: Evidence from CFPS. Economic Modelling, 2020, 90, 386-396.
- Zhang, H.; Wang, H.Y.; Yan, H.Y.; Wang, X.Y. Impact of Internet Use on Mental Health among Elderly Individuals: A Difference-in-Differences Study Based on 2016-2018 CFPS Data. International Journal Of Environmental Research And Public Health 2022, 19.
“Lines 126-132: A more detailed description of the used questionnaires would be helpful, including type of instruments, psychometric properties, and few item examples. Also, the rating of responses and total score calculation are not clear. Please be more specific regarding these aspects.” It is not only about the description of items/ content of the scale. Research on the psychometric properties of this questionnaire are necessary (reliability, validity, etc).
Among all the variables used in this study, except the level of depression and the perceived wealth gap, other variables are demographic characteristics and belong to objective facts that can be measured directly.
The CFPS questionnaire used the center for emotional Studies Depression Scale (CES-D) to test the individual's depression level. The CES-D scale is a set of mature, standardized and widely used measurement tools in academic circles. It was found to have very high internal consistency and adequate testretest repeatability. Validity was established by patterns of correlations with other self-report measures, by correlations with clinical ratings of depression, and by relationships with other variables which support its construct validity. Reliability, validity, and factor structure were similar across a wide variety of demographic characteristics in the general population samples tested. The scale should be a useful tool for epidemiologic studies of depression [5].
CFPS2018 questionnaire adopts the 8-question version of ces-d8, and the scores of ces-d8 and ces-d20 are equivalently operated through the equipercentile equating. In other words, both the total score of 8-question version and the score of cesd20sc are adopted in the questionnaire. In this article, we use the score of cesd20sc.
The perceived wealth gap is measured by a question: in general, how serious do you think the gap between the rich and the poor is in China (0 means not serious, 10 means very serious)? Respondents will choose a number from 0 to 10 to express their attitude. When dealing with variables, we recoded 0 to 5 as 0, indicating that the respondents believe that it is not serious. The code of 6 to 10 is 1, which means that the respondents believe that it is serious.
References:
[5]Radloff, L. S. The CES-D Scale: a Self-Report Depression Scale for Research in the General Population. Applied Psychological Measurement 1977, 1, 385-401.
https://journals.sagepub.com/doi/10.1177/014662167700100306
- Conclusions and Discussions
A general conclusion should be drawn upon the entire study.
added:
- Conclusions
The results of this study show that the use of mobile Internet can effectively reduce the prevalence of depression of Chinese young adults. This effect was significant in people with higher levels of depression, older age, and who use Internet for communicative purpose. However, there is no significant difference in mobile Internet use between the sexes. In addition, the use of mobile Internet can improve young people's sense of social comparison, leading to a decrease in depressive feelings. Our findings help to improve the public and government's understanding of depression in young adults, and provide suggestions for the prevention and relief of depression.
Please include the definition of netizens in the manuscript.
Added: Netizens, who conduct online activities through computers and the Internet,
“Statements about funding, acknowledgement, data availability and conflict of interest are missing.”
Author Contributions: Yuyan Chen: conceptualisation, methodology, software, formal analysis, data curation, writing. Lin Wu: validation, investigation, resources, and writing—review and editing. Zenghua Guo: conceptualization, methodology, writing—original draft preparation, supervision, project administration and funding acquisition of the study. All authors have read and agreed to the published version of the manuscript.
Conflict of Interest: The authors declare that the research was conducted in the absence of any commercial or financial relationships that could be construed as a potential conflict of interest.
Data availability:This paper uses data from the Chinese Family Panel Studies (CFPS), a nationally representative longitudinal survey funded by Peking University and the National Natural Science Foundation of China and initiated by the Institute of Social Sciences of Peking University
FUNDING
This research was funded by the “Major Program of National Fund of Philosophy and Social Science of China, Grant Number 18VZL009.”
Authors mentioned that the research is funded by Peking University and the National Natural Science Foundation of China (answers). This information should be disclosed.
This information is about CFPS, not this study.
In addition, some mistakes in expression and spelling have been corrected.
